# Certified algorithms for equilibrium states of local quantum Hamiltonians

Hamza Fawzi[1], Omar Fawzi [2] & Samuel O. Scalet [1] ✉

Predicting observables in equilibrium states is a central yet notoriously hard question in quantum many-body systems. In the physically relevant thermodynamic limit, certain mathematical formulations of this task have even been shown to result in undecidable problems. Using a finite-size scaling of algorithms devised for finite systems often fails due to the lack of certified convergence bounds for this limit. In this work, we design certified algorithms for computing expectation values of observables in the equilibrium states of local quantum Hamiltonians, both at zero and positive temperature. Importantly, our algorithms output rigorous lower and upper bounds on these values. This allows us to show that expectation values of local observables can be approximated in finite time, contrasting related undecidability results. When the Hamiltonian is commuting on a 2-dimensional lattice, we prove fast convergence of the hierarchy at high temperature and as a result for a desired precision $\varepsilon$, local observables can be approximated by a convex optimization program of quasi-polynomial size in $1/\varepsilon$.

A central question in physics is to determine the properties of a many-body quantum system as a function of the interaction between its constituents. The topic of Hamiltonian complexity[1,2] studies this question from a complexity-theoretic point of view. The results in Hamiltonian complexity suggest that efficient algorithms answering this question are unlikely to exist. In fact, determining the ground energy is hard for the complexity class QMA[3], and this even holds for translation-invariant systems[4]. This means that we do not expect polynomial-time (classical or quantum) algorithms for this problem. Computing expectation values of observables in the ground state is even harder than the ground energy[5]. In the thermodynamic limit, when the number of systems is taken to infinity, finding good approximations to the energy can be hard for fixed Hamiltonians[6,7], and the spectral gap is even uncomputable[8].

These complex results put severe limitations on provably efficient and correct classical and quantum algorithms for the fundamental questions in many-body physics. However, these limitations only apply to algorithms that have the required properties for all valid instances of the problem. Moreover, typically the instances showing hardness are highly contrived. In order to avoid these limitations, we consider here certified algorithms where we require the correctness condition

for all instances, but relax the condition of provable efficiency for all instances. We say that an algorithm for computing $p^*(H)$ is *certified* if on input $H$, it outputs a pair of numbers $(p_\ell^{min}, p_\ell^{max})$ such that we have $p^*(H) \in [p_\ell^{min}, p_\ell^{max}]$ for all $H$ and all $\ell$, i.e., the algorithm provides upper and lower bounds on the quantity of interest. Even without having any performance guarantees a priori, the error is bounded by the size of the interval a posteriori.

We further require that as $\ell \to \infty$, the interval $[p_\ell^{min}, p_\ell^{max}]$ converges to the desired value $p^*(H)$. Here, $\ell$ is a parameter that governs the runtime of the algorithm such that fast convergence in $\ell$ leads to an efficient algorithm. The above cited complexity results show, e.g., that if $p^*(H)$ is the ground energy of $H$, then for small $\ell$, the interval $[p_\ell^{min}, p_\ell^{max}]$ has to be large for some $H$. However, for any input $H$ of interest, we can always run the algorithm and the returned information will be correct and certifiably so. As such, no a priori analysis of the convergence speed is needed in order to obtain rigorous approximations of the quantity of interest.

A well-studied general way of determining properties of thermal states is by preparing such states on a quantum computer. This includes, for example, works on quantum Metropolis sampling[9,10] and, more generally, quantum Gibbs samplers[11,12]. Here, we focus on

[1]Department of Applied Mathematics and Theoretical Physics, University of Cambridge, Cambridge, United Kingdom. [2]Univ Lyon, Inria, ENS Lyon, UCBL, LIP, Lyon, France. ✉e-mail: sos25@cam.ac.uk

*classical* algorithms, which clearly cannot prepare quantum thermal states, but can still compute specified properties of general quantum thermal states. We note that our algorithms have a similar feature to (quantum) Metropolis sampling and, more generally, (quantum) Markov Chain Monte Carlo methods, in that they can be applied for any local Hamiltonian, but this comes at the (unavoidable) cost of not always having fast convergence.

In this work, we consider a hierarchy of convex optimization problems that provide certified algorithms for computing observables in ground and thermal states in finite and infinite systems. As a main result, we prove the decidability of a formulation of the equilibrium observable problem, which contrasts with previous closely related undecidability results. The main technical ingredient is a formulation of constraints relaxing the Gibbs condition in a convex and algorithmically feasible manner. Furthermore, we are able to prove efficiency in a more restricted setting. Preliminary numerical results also suggest that the method can be of use in practice.

## Results

We now present the setup in an informal way. We refer to the Supplementary information for more formal statements. We assume that our system is described by a local Hamiltonian on a discrete set of sites $\Gamma$, which can be formally written as

$$H = \sum_{X \subset \Gamma} h_X \tag{1}$$

where $h_X$ is the interaction term acting only on the finite set of sites $X \subset \Gamma$. We assume that the Hilbert space describing each site has finite dimension $d$ and the Hamiltonian $H$ is local, meaning that $h_X$ is nonzero only when $X$ has a size smaller than a constant. For finite systems, the set of equilibrium states at temperature $T \geq 0$ is the set of states $\rho$ that minimize $\text{tr}(H\rho) - TS(\rho)$, where $\text{tr}(H\rho)$ is the energy, and $S(\rho)$ is the entropy. At $T > 0$, this set reduces to a single equilibrium state, also called the Gibbs state given by $\rho = \frac{e^{-H/T}}{\text{tr} e^{-H/T}}$. However, for infinite systems, the thermal equilibrium state is not necessarily unique, and this property lies at the heart of the existence of phase transitions.

Given the description of such a system, our objective is to determine the physical properties of the corresponding equilibrium states. Consider a local observable $O$, e.g., the magnetization in the z-direction for a spin-1/2 system for some site $x \in \Gamma$. We are interested in the set of values $\text{tr}(\rho O)$ that an equilibrium state at temperature $T = 1/\beta$ can take:

$$[\langle O \rangle_\beta^{\min}, \langle O \rangle_\beta^{\max}] := \{\text{tr}(\rho O) : \rho \text{ is an equilibrium state at temperature } T = 1/\beta\}. \tag{2}$$

Note that for a finite system and $\beta < \infty$, we have $\langle O \rangle_\beta^{\min} = \langle O \rangle_\beta^{\max}$ as the thermal state is unique. However, we can have $\langle O \rangle_\beta^{\min} < \langle O \rangle_\beta^{\max}$ for $\beta = \infty$ if the ground space of $H$ is degenerate and for infinite systems when there are multiple thermal equilibrium states.

### Main result

We formulate a hierarchy of convex optimization programs which produce converging lower (resp. upper) bounds on $\langle O \rangle_\beta^{\min}$ (resp. $\langle O \rangle_\beta^{\max}$) for any $\beta \geq 0$. Our optimization problems are formulated in terms of the matrix-valued relative entropy function

$$D_{op}(A \| B) = A^{1/2} \log(A^{1/2} B^{-1} A^{1/2}) A^{1/2} \tag{3}$$

defined for arbitrary positive definite matrices $A$, $B$, and which is jointly convex in $(A, B)$. Consider a finite subset $\Lambda$ of the lattice sites $\Gamma$ containing the support of the local observable $O$. Then we can formulate the following convex optimization program over density operators supported on $\overline{\Lambda} = \Lambda \cup \partial^{ex}\Lambda$ where $\partial^{ex}\Lambda$ is the external

boundary of $\Lambda$ (see the Supplementary information Equation (4) for the precise definition):

$$\min_\rho / \max \quad \text{tr}(\rho O) \tag{4}$$

$$\text{s.t.} \quad \rho \text{ density operator on } (\mathbb{C}^d)^{\otimes |\overline{\Lambda}|} \tag{5}$$

$$\text{tr}_{\overline{\Lambda} \setminus \Lambda}(H_{\overline{\Lambda}} \rho - \rho H_\Lambda) = 0 \tag{6}$$

$$D_{op}\left(A_\rho \| B_\rho\right) \preceq \beta C_\rho. \tag{7}$$

Here, $H_{\overline{\Lambda}}$ is the truncated Hamiltonian acting on $\overline{\Lambda}$, and $A_\rho$, $B_\rho$, $C_\rho$ are Hermitian matrices that depend linearly on $\rho$, defined by:

$$(A_\rho)_{ij} = \text{tr}(\rho a_i^* a_j), \quad (B_\rho)_{ij} = \text{tr}(\rho a_j a_i^*), \quad (C_\rho)_{ij} = \text{tr}(\rho a_i^*[H_{\overline{\Lambda}}, a_j]) \tag{8}$$

where $\{a_i\}$ is a basis of the space of operators acting on $(\mathbb{C}^d)^{\otimes |\overline{\Lambda}|}$. Note that the program (4-7) involves a matrix variable of dimension $d^{|\overline{\Lambda}|} \times d^{|\overline{\Lambda}|}$ and convex constraints involving matrices of dimension at most $d^{2|\overline{\Lambda}|} \times d^{2|\overline{\Lambda}|}$.

By taking larger and larger subsets $\Lambda \uparrow \Gamma$, one can prove that the solutions of the convex optimization programs will converge to the expectation values $\langle O \rangle_\beta^{\min}$ and $\langle O \rangle_\beta^{\max}$. This is the content of the following theorem.

**Theorem 2.1.** (Certified algorithms for expectation values of equilibrium states). Let $\Lambda_0 \subset \Lambda_1 \subset \cdots \subset \Gamma$ be an increasing sequence such that the support of the local observable $O$ is contained in $\Lambda_0$. For any $\ell \in \mathbb{N}$, let $p_\ell^{\min}$ and $p_\ell^{\max}$ be respectively the minimum and maximum values of the convex optimization problems (4-7) with $\Lambda = \Lambda_\ell$. Then we have

$$p_\ell^{\min} \leq \langle O \rangle_\beta^{\min} \leq \langle O \rangle_\beta^{\max} \leq p_\ell^{\max}. \tag{9}$$

Furthermore, $p_\ell^{\min} \uparrow \langle O \rangle_\beta^{\min}$ and $p_\ell^{\max} \downarrow \langle O \rangle_\beta^{\max}$ as $\ell \to \infty$.

We note that for finite systems, convergence happens after a finite number of steps, namely for $\ell$ such that $\Lambda_\ell = \Gamma$. While the convergence result becomes trivial in the finite case, the formulation of a convex program containing constraints for the Gibbs state still adds a novel tool for the numerical treatment of finite-sized quantum many-body systems. The more interesting case for our results is however that of infinite systems. Given a fixed computational budget, one can choose $\ell$ and run the corresponding program (4-7) and obtain a superset $[p_\ell^{\min}, p_\ell^{\max}]$ of the target interval $[\langle O \rangle_\beta^{\min}, \langle O \rangle_\beta^{\max}]$. In the cases where $\langle O \rangle_\beta^{\min} = \langle O \rangle_\beta^{\max}$, e.g., for finite systems at $T > 0$, one can also run the program (4-7) with increasing values of $\ell$ until $p_\ell^{\max} \leq p_\ell^{\min} + \varepsilon$, where $\varepsilon$ is some desired accuracy $\varepsilon$. We note that, as previously mentioned, the problem of computing expectation values for equilibrium states is at least QMA-hard so we cannot hope to have fast convergence for all choices of $H$. But we stress that no a priori analysis of convergence speed is required to obtain some desired accuracy: as soon as we get $p_\ell^{\max} \leq p_\ell^{\min} + \varepsilon$, an additive $\varepsilon$ approximation is guaranteed for this instance. Numerical results illustrating this algorithm can be found in the Supplementary information Section 3.

### Translation-invariant infinite systems

Translation-invariant Hamiltonians on the infinite lattice $\Gamma = \mathbb{Z}^D$, i.e., satisfying $h_X = h_{X+x}$ for all $x \in \mathbb{Z}^D$, play a specifically important role in statistical physics, in particular for understanding phase transitions. For such systems, one often considers the expectation value per site of an observable, for example, the energy per particle also called the energy density. One way to define expectation values per site is to compute the observable $O$ on any fixed site on a translation-invariant

## BOX 1

# Equilibrium Observable Problem

**Input:** Local dimension $d$, local Hamiltonian term $h$ with rational coefficients, local observable $O$ with rational coefficients, rational number $a$, temperature $\beta \in [0, +\infty]$

**Promise:** Either $\langle \mathbf{O} \rangle_\beta^{\min,\mathrm{TI}} > \mathbf{a}$ or $\langle \mathbf{O} \rangle_\beta^{\max,\mathrm{TI}} < \mathbf{a}$

**Question:** Output YES if $\langle \mathbf{O} \rangle_\beta^{\min,\mathrm{TI}} > \mathbf{a}$, NO otherwise

equilibrium state of $H$. More precisely, we define the average expectation value of observable $O$ per site to be the interval

$$[\langle O \rangle_\beta^{\min,\mathrm{TI}}, \langle O \rangle_\beta^{\max,\mathrm{TI}}]:$$
$$:= \{ \mathrm{tr}(\rho O) : \rho \text{ is a translation-invariant equilibrium state at temperature } T = 1/\beta \}. \quad (10)$$

By adding to the program (4-7) a translation-invariance constraint in $\overline{\Lambda}$ given by

$$\mathrm{tr}_{\Sigma^c}(\rho) = \mathrm{tr}_{(\Sigma+x)^c}(\rho) \qquad \text{for all } \Sigma \subset \overline{\Lambda}, x \in \mathbb{Z}^d \text{ such that } \Sigma + x \subset \overline{\Lambda}, \quad (11)$$

Theorem 2.1 can be adapted for translation-invariant states, as follows:

**Theorem 2.2.** Let $\Lambda_0 \subset \Lambda_1 \subset \cdots \subset \Gamma$ be an increasing sequence such that the support of the local observable $O$ is contained in $\Lambda_0$. For any $\ell \in \mathbb{N}$, let $\mathsf{p}_\ell^{\min,\mathrm{TI}}$ and $\mathsf{p}_\ell^{\max,\mathrm{TI}}$ be respectively the minimum and maximum values of the convex optimization problems given by (4-7) together with the constraint (11). Then we have

$$\mathsf{p}_\ell^{\min,\mathrm{TI}} \leq \langle O \rangle_\beta^{\min,\mathrm{TI}} \leq \langle O \rangle_\beta^{\max,\mathrm{TI}} \leq \mathsf{p}_\ell^{\max,\mathrm{TI}}. \quad (12)$$

Furthermore, $\mathsf{p}_\ell^{\min,\mathrm{TI}} \uparrow \langle O \rangle_\beta^{\min,\mathrm{TI}}$ and $\mathsf{p}_\ell^{\max,\mathrm{TI}} \downarrow \langle O \rangle_\beta^{\max,\mathrm{TI}}$ as $\ell \to \infty$.

One difficulty we would like to highlight regarding the thermodynamic limit concerns the definition of equilibrium states at any given temperature $T \geq 0$. It was actually shown in ref. 13 that computing local observables on the ground state of infinite systems is undecidable. This might seem to contradict our result. This is not the case, however, due to a subtle point in the definition of ground-state observables in the infinite limit. The authors of ref. 13 define those as limits of ground-state observables in finite systems, i.e., the observable is computed for the ground-state of a truncated Hamiltonian with open boundary condition $H_\Lambda = \sum_{X \subset \Lambda} h_X$, where $\Lambda \subset \mathbb{Z}^D$ is finite, and afterward the limit $\Lambda \uparrow \mathbb{Z}^D$ is taken. The problem definition we use in our work, which is standard in the operator algebraic framework, is more general and, in particular, contains limits of ground-state observables of finite system Hamiltonians with any choice of boundary conditions. To avoid confusion, it should be added that this does not mean that our algorithm can compute the value of an observable for a specific boundary condition. Instead, it gives an outer relaxation of the interval given by all boundary conditions. A boundary condition is not input to the algorithm. It is simple to construct a Hamiltonian where fixing the boundary condition while taking the limit excludes natural ground states (see Remark 1.1 and Remark 2.3 in the supplemental information concerning thermal states of the 2D Ising model). The comparison, however, illustrates that even the existence of any convergent algorithm for observables in ground states is far from obvious, which is what we achieve in Theorem 2.2. Considering the operator algebraic definition of equilibrium states leads to the natural computational problem defined in Box 1.

Note that this is a promise problem where the objective is to decide if all the equilibrium states have an expectation value per site $> a$ or all the equilibrium states have an expectation value per site $< a$.

**Theorem 2.3.** (Decidability of translation-invariant ground state and thermal observables). The equilibrium observable problem is decidable.

**Main ingredients**

For clarity of the discussion, we focus here on finite systems. A key ingredient to obtain our algorithms is to use an operator algebraic characterization of equilibrium states. These are expressed solely in terms of $\mathrm{tr}(\rho b)$ for operators $b$. For $T = 0$, the common definition of an equilibrium state is a state supported on the eigenspace of $H$ with the minimum eigenvalue. It turns out that an equivalent operator algebraic formulation of this condition is:

$$\mathrm{tr}(\rho a^*[H,a]) \geq 0 \ \forall a \quad (13)$$

where $[H, a]$ is the commutator and $a$ is an arbitrary observable. Intuitively, the condition above expresses the fact that the energy of $\rho$ has to increase under any perturbation. There are two crucial facts about (13): First, if $a$ is supported on a small set $\Lambda$ of sites then the condition (13) only depends on $\mathrm{tr}(\rho b)$ for operators $b \in \overline{\Lambda}$, where we recall that $\overline{\Lambda} = \Lambda \cup \partial^{ex}\Lambda$. Second, the inequalities in (13) can be concisely captured by a convex constraint involving the positivity of a Hermitian matrix that depends linearly on $\rho$. By restricting the operators $a$ in (13) to be supported on $\Lambda$, this leads to (4-7) for $T = 0$.

At positive temperature $T = 1/\beta > 0$, the situation is more complicated. As previously mentioned, the common definition of a thermal equilibrium state is given by $\rho = \frac{e^{-\beta H}}{\mathrm{tr}(e^{-\beta H})}$. It turns out that an equivalent operator algebraic formulation is via the Kubo–Martin–Schwinger (KMS) condition:

$$\mathrm{tr}(\rho ba) = \mathrm{tr}(\rho a e^{-\beta H} b e^{\beta H}) \quad \forall a, b. \quad (14)$$

In the classical case, when the Hamiltonian is diagonal in a basis $\{|\sigma\rangle\}_\sigma$, Equation (14) reduces to

$$\mathrm{tr}(\rho |\sigma\rangle\langle\sigma|) = \mathrm{tr}(\rho |\sigma'\rangle\langle\sigma'|) \exp(-\beta(\langle\sigma|H|\sigma\rangle - \langle\sigma'|H|\sigma'\rangle)), \quad (15)$$

for all $\sigma, \sigma'$. When $\sigma'$ is obtained from $\sigma$ by flipping a single spin, these are known as the spin-flip equations and are exploited in the bootstrap approach for the classical Ising model[14].

To obtain Theorem 2.1, a natural idea (as was done for $T = 0$) is to relax the set of $\beta$-KMS states and impose the condition (14) for a subset of observables $a$, $b$ supported on some small $\Lambda$. However, the main obstruction one is faced with is that even if $a$, $b$ are local observables, the expression $ae^{-\beta H}be^{\beta H}$ is, in general, not local, except for commuting Hamiltonians. As such, even though (14) forms a set of linear equations on $\rho$, they involve the expectation of $\rho$ on nonlocal observables. We circumvent this issue by using another characterization of thermal equilibrium states via so-called energy-entropy balance (EEB) inequalities[15]: this is an infinite set of scalar convex inequalities, each indexed by an operator $a$, which carve out the set of $\beta$-KMS states. On the one hand, these inequalities are better suited than the $\beta$-KMS condition because they preserve locality, i.e., they only require the expectation value of the state $\rho$ on a finite region around the support

of $a$. On the other hand, a drawback of these inequalities is that there are infinitely many of them, and unlike the inequalities (13), it is not possible to express them as a linear positive semidefinite constraint. A key ingredient to prove our theorem is to formulate a matrix generalization of such inequalities using the matrix-valued relative entropy function (3). We show that an infinite set of scalar energy-entropy balance inequalities can be compactly formulated by a single nonlinear convex matrix inequality of the form $D_{op}(A_\rho \| B_\rho) \preceq \beta C_\rho$ for some suitable matrices $A_\rho, B_\rho, C_\rho$ that depend linearly on $\rho$ (see (7)). To the best of our knowledge, this is the first application of the matrix relative entropy function in the design of convex relaxations in quantum information. Optimization problems involving this function can be solved using interior-point methods[16] or via semidefinite programming approximations[17].

### Quantifying convergence speed

Theorems 2.1 and 2.2 do not provide quantitative guarantees on the convergence speed. As previously mentioned, one cannot hope to prove general fast convergence guarantees as even the special case where the observable $O$ corresponds to energy is unlikely to have a polynomial-time quantum algorithm, even when $D = 1$[4].

However, one can expect provably fast convergence for some classes of Hamiltonians. We illustrate this by showing exponential convergence in two regimes for which it is known that no phase transitions can occur. For translation-invariant Hamiltonians in the high-temperature regime and in one dimension at any nonzero temperature, the set of equilibrium states reduces to a singleton, and for any local observable $O$ we have $\langle O \rangle_\beta^{\min,\mathrm{TI}} = \langle O \rangle_\beta^{\max,\mathrm{TI}} = \langle O \rangle_\beta^{\mathrm{TI}}$. Assuming a commuting local Hamiltonian $H$, the theorem below shows exponential convergence to $\langle O \rangle_\beta^{\mathrm{TI}}$ in the level $\ell$ of the convex optimization hierarchy.

**Theorem 2.4.** (Quantitative convergence rate). Consider a translation-invariant local Hamiltonian $H$ on $\Gamma = \mathbb{Z}^D$ with $D \leq 2$. Assume furthermore that $H$ is *commuting*, i.e., $[h_X, h_Y] = 0$ for all $X, Y \subset \mathbb{Z}^D$. For $D = 1$ and $\beta_1 = \infty$ or for $D = 2$ and some $\beta_1 > 0$, we have for all $0 \leq \beta < \beta_1$, and for any local observable $O$, $\langle O \rangle_\beta^{\min,\mathrm{TI}} = \langle O \rangle_\beta^{\max,\mathrm{TI}} = \langle O \rangle_\beta^{\mathrm{TI}}$ and, using the same notation as in Theorem 2.1

$$\max\{\mathsf{p}_\ell^{\max} - \langle O \rangle_\beta^{\mathrm{TI}}, \langle O \rangle_\beta^{\mathrm{TI}} - \mathsf{p}_\ell^{\min}\} \leq c_1 \| O \| e^{-c_2 \ell} \qquad (16)$$

for some constants $c_1, c_2 > 0$ depending on the dimension and the interaction.

This result addresses an open problem raised in[18] about the speed of convex optimization hierarchies for the (classical) Ising model, in particular, whether exponential convergence holds away from criticality. Theorem 2.4 establishes such a statement for the more general class of commuting local Hamiltonians.

## Discussion

A special case of the problem considered in this work is when the observable $O$ is the energy. Certified algorithms do exist for the ground energy of local Hamiltonians: one can combine semidefinite programming relaxations[19–22] for lower bounds and variational methods such as tensor networks[23–25] for upper bounds. Using such two-sided bounds on the ground energy, the recent works[26,27] obtain bounds for expectation values of local observables in the ground state, although no convergence guarantees are given. The approach based on imposing the additional constraint (13), which leads to convergence guarantees, was proposed independently and concurrently in[28], where it was derived as a special case of a method to strengthen semidefinite relaxations for noncommutative polynomial optimization problems by incorporating optimality conditions as additional constraints. These papers, however, do not discuss the case of positive temperature since the equilibrium states are not characterized by a noncommutative polynomial optimization problem which is linear in the state. One could try to use lower

bounds for the free energy coming from convex relaxations[29], but it is not clear how to use such bounds for observables. We note, however, that for classical systems, thermal observables can be obtained via the bootstrap approach[14,30], by directly imposing the KMS conditions (15). The resulting hierarchies were shown to be asymptotically convergent[18], even if only a weaker set of constraints are imposed. For quantum systems, to the best of our knowledge, Theorem 2.1 provides the first certified algorithms for general observables in thermal states.

Note that, in some restricted settings, such as 1D gapped systems[31] at zero temperature or 1D systems at positive temperature[32–35], there are provably efficient algorithms computing representations of equilibrium states and thus expectation values, but such algorithms are tailored to these settings. In addition, for arbitrary dimensions and high temperatures, provably efficient algorithms for computing the free energy exist[36–38], which can be used to compute observables[39].

The preliminary numerical experiments in the supplemental information demonstrate that the approach proposed in this paper is not only theoretical, and with additional efforts on the computational side, can play an important role alongside other classical algorithms. For example, because our algorithm produces certified bounds, it can be used to rigorously benchmark variational algorithms for quantum many-body systems[40]. In addition, one can easily identify several directions for improving the accuracy of the algorithm presented here: First, one can exploit additional convex constraints that are known to hold for marginals of equilibrium states. For example, one can add entropy-based inequalities as proposed in ref. [41]. Furthermore, other valid inequalities tailored to specific models can be added to the convex relaxation such as reflection positivity which was used in ref. [14] for the Ising model. Second, it would be very interesting to combine the methods from this paper with variational methods (e.g., from tensor networks) to obtain more accurate bounds, such as the recent work on the ground energy problem[42]. Another natural question is whether one can use our algorithms within hybrid classical-quantum algorithms for quantum simulation problems[43]. On the analytical side, it remains open whether convergence guarantees for other classes of Hamiltonian can be proven. Promising candidates would be regimes in which other classical algorithms are efficient such as thermal states in 1D and at high-temperature (for general noncommuting Hamiltonians) as well as gapped ground states in 1D.

## Data availability

No datasets were generated or analysed during the current study.

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

## Acknowledgements

We thank Daniel Stilck França for the helpful discussions. H.F. acknowledges funding from UK Research and Innovation (UKRI) under the UK government's Horizon Europe funding guarantee EP/X032051/1. O.F. acknowledges funding by the European Research Council (ERC Grant AlgoQIP, Agreement No. 851716) as well as by the European Union's Horizon 2020 within the QuantERA II Program under Grant VERIqTAS Agreement No 101017733. S.O.S. acknowledges support from the UK Engineering and Physical Sciences Research Council (EPSRC) under grant number EP/W524141/1.

## Author contributions

H.F., O.F., and S.O.S. contributed equally to this work. H.F., O.F., and S.O.S. contributed to discussions, proofs, and preparation of the manuscript.

## Competing interests

The authors declare no competing interest.
