## [Peer Review File · Nature Communications]

Certified algorithms for equilibrium states of local quantum HamiltoniansREVIEWER COMMENTS

Reviewer #1 (Remarks to the Author):

Review of NCOMMS-24-13405-T

In recent years it has been shown that the complexity of behaviour exhibited by complex quantum many-body systems allows to find numerous instances of undecidable properties, i.e., physical properties which cannot be predicted by any algorithm (no matter the running time). Very briefly, this is due to the fact that the complexity of these systems is sufficient to represent undecidable problems in them, such as the Halting problem. An active area of research has been therefore to understand where is the threshold between decidable and undecidable properties, since only the former are amenable to be treated by numerical analysis.

This manuscript presents an algorithm to upper and lower bound the possible expectation values of observables in the equilibrium states of local quantum Hamiltonians in the thermodynamic limit (which covers thermal equilibrium states at finite temperature and ground states at zero temperature). The algorithm has no guarantee on the running time, although the authors expect (and provide some numerical evidence) that it could also be relatively efficient in some instances. But the central novelty of the work is that it shows that the algorithm is guaranteed to converge, and therefore the problem considered is decidable.

This result is particularly interesting as it contrasts with previous works, which have shown that very similar problems are instead undecidable. The relation to [BCW21] is of particular relevance. In [BCW21], it was shown that computing expectation values of local observables on ground states is undecidable, which seems to contradict the result in the current work. The difference between the two results is crucially in how the problem is defined. In the work of [BCW21], they consider systems defined on large but finite volumes with open boundary conditions, and the problem is to decide whether the expectation value of a local observable is eventually 0 or 1, as the system size is scaled to infinity. In the current manuscript, the problem is instead to find the minimal/maximal value of the expectation value with respect to the full set of ground states in the thermodynamic limit (i.e. in the infinite volume). These ground states can be thought as limits of finite volumes with different boundary conditions, so that the comparison between the two works could be summarized as follows:

- If one is interested in the expectation value of a local observable with a specific choice of boundary conditions, the problem is undecidable, according to [BCW21];
- If one is instead interested in obtaining the minimum/maximum expectation value of a local observable, over all the possible choices of boundary conditions, then the algorithm presented in the current manuscript shows that the problem is decidable.

This is quite a surprising result, and we believe this makes the paper worthy of publication. There are few results in the literature that show that certain properties of infinite systems are decidable (most of the famous ones show undecidability under certain conditions), and they are usually very restrictive in their conditions (e.g., 1D lattices or qubits/two-level systems).

From the point of view of the presentation, the manuscript does a very effective job at introducing and explaining the C^* -algebra formulation of infinite volumes ground states, from which the convex relaxation on which the algorithm is based. The paper is carefully built up and thoroughly explained, therefore being easy to read. All necessary proofs were detailed, as well as needed references to previous literature.

Nonetheless, there are two crucial points which we believe the paper does not do a good job at explaining, and which we think should be improved before the paper is published.

1. Infinite ground states and boundary conditions

In Remark 2.1, it is explained that limits of finite volumes ground states produce ground states in the thermodynamic limit, so that different infinite volumes ground states can occur due to different choices of boundary conditions. The latter are defined as extra Hamiltonian terms Δ_{Λ^c} acting on the “external boundary” of a finite volume Λ . Unfortunately the example that follows does not really exemplify what has been just explained. While it contains an example of two different infinite volume ground states, it does not show (at least not explicitly) how these arise from different choices of boundary conditions. We believe making this explicit could be very useful to the readers.

Moreover, we think that there are other choices of examples that could have worked better and are more widely known, which we think could be included in addition or substitution of the current one. One example is of course the Ising model, either in 1D or 2D, which is then used as a numerical example in Section 4. Another good example could be Kitaev’s toric/planar code, which has single-particle ground states in the limit [CNN18] (which can be obtained by fixing certain boundary conditions).

More generally, since this is a crucial point which separates the current work from the previous results in the literature, this part should be extended and better clarified.

2. The present algorithm applied to the model on [BCW21]

We think that the discussion at the beginning of section “3.4 Decidability”, regarding the relationship between the current work and [BCW21], should be improved and expanded. One question which is not clear from that section is: what happens if we apply the algorithm presented in this work to the family of models constructed in [BCW21]?

We intuitively believe that the answer should be that the minimum and maximum expectation value of the local observable are always the same, and thus do not depend on whether the particular instance of the problem corresponds to a halting/non-halting machine. Is this interpretation correct? If so, it should be properly explained. Moreover, given the explanation that different infinite volume ground states arise as limits of finite volumes with different boundary conditions, is it possible to explicitly construct the two boundary conditions that realize the minimum/maximum value for these family of models? I.e., can we explicitly construct which boundary conditions we need to impose on the models from [BCW21] that make the problem decidable? Or is the current result showing that such boundary conditions exists, but it could be that constructing them explicitly is also an undecidable problem?

References

[BCW21] Johannes Bausch, Toby S. Cubitt, and James D. Watson. Uncomputability of phase diagrams. *Nature Communications*, 12(1), January 2021. URL: <http://dx.doi.org/10.1038/s41467-020-20504-6>, doi:10.1038/s41467-020-20504-6.

[CNN18] M. Cha, P. Naaijkens, and B. Nachtergaele. The Complete Set of Infinite Volume Ground States for Kitaev’s Abelian Quantum Double Models. *Commun. Math. Phys.* 357, 125–157 (2018). <https://doi.org/10.1007/s00220-017-2989-4>

Reviewer #2 (Remarks to the Author):

Reviewer #3 (Remarks to the Author):

This paper introduces a new family of algorithms based on convex optimization to compute equilibrium expectation values for quantum Hamiltonian systems. Equilibrium in this case refers to either the ground state or the Gibbs state at some finite temperature. While such problems naturally have variational formulations, these do not necessarily yield tractable algorithms. This paper is mainly concerned with quantum spin systems in the thermodynamic limit of infinite number of spins where the issue of tractability is more acute. An avenue for dealing with this is to use so-called convex relaxations which approximate the original quantity via easier convex optimization problems by e.g., choosing an objective that is easier to compute or removing some of the constraints in the original optimization.

The authors use operator algebraic characterization of equilibrium quantum observables and quantum Shannon theory tools to formulate a (possibly infinite) hierarchy of convex relaxations to the problem of computing equilibrium observables. They prove that their algorithms converge to the correct answer as one goes to higher levels of the hierarchy, i.e., they are certified. However, there is no guarantee on the time to convergence in the general case. In special instances, specifically for 1D commuting Hamiltonians and more general 1D Hamiltonians at sufficiently high temperatures, they are able to prove that the algorithms converge exponentially fast. As per my understanding, classical algorithms with similar running-times are already known to exist for these instances by other techniques. So the results here do not seem to give new families of efficiently solvable quantum systems. Nevertheless, these are interesting from the perspective of developing new tools for tackling quantum many-body problems.

Even in the absence of convergence-rate bounds, the fact that the algorithms converge have

some bearing on questions about decidability of quantum many-body problems. It was shown previously in a paper by Bausch et al., that the problem of deciding whether a ground-state observable is above or below a threshold for spin chains in the thermodynamic limit is formally undecidable. That the algorithms in this paper are certified appear to show, on the other hand, that this problem is decidable. The authors comment on this contradiction -- the resolution is that the operator algebraic formulation leads to a different problem in the thermodynamic limit. The reading of the text suggests to me that the latter is physically better motivated.

The authors also perform numerical experiments on the 1D transverse-field Ising model to demonstrate their algorithm. The results show reasonable agreement with the exact values at low levels of the hierarchy.

The main contribution in the paper is the formulation of the relaxation for infinite-size systems. For this, the authors use a characterization of thermal equilibrium states known as the energy-entropy balance inequalities (EEB) which give a set of constraints on the moments of observables in the thermal state. The set of these inequalities is infinite and the relaxation is to enforce inequalities corresponding to observables in a subalgebra. The authors show how the EEB inequalities can be formulated as an inequality involving a matrix-valued relative entropy. This allows them to use prior work on semidefinite approximations to quantum entropies to write the final optimization as a semidefinite program (SDP). Known bounds on convergence of SDPs along with other results then imply the certifiability of their algorithms and the fast convergence in special cases.

Overall, I find this to be a very nice paper building interesting new tools in the arena of classical algorithms for many-body systems. I believe that there is substantial promise for future work building upon the results here.

The exposition is very clear given the technicality of the content. The review of background material, e.g., the discussion about algebraic characterization of equilibrium states may be broadly useful. However, it does require some familiarity with the C^* -algebraic language. Thus it might help to include some basic definitions for a broader audience - what is a state,

or what is meant by "extendible", etc.

I am happy to recommend publication of this article once the authors address a few minor comments below.

Other comments:

1. Can you comment on the applicability of your algorithms to finite-sized but large systems? Do these give any new tools or ideas in that setting?
2. Page 10, Section 3.2: Can you elaborate why eq. 17e becomes an infinite list of scalar inequalities whereas eq. 17c is not? Related to the previous question, what happens when the system is finite-sized? Can I get a nicer set of inequalities by some kind of polynomial approximation to the logarithm?
3. Same section, "...outer relaxation for the set of λ -marginals..": Here it may be helpful to emphasise that the relaxation is still optimising over states. This threw me off a little since other outer approximations in the literature do not necessarily optimise over actual states.
4. I was wondering if SDPs can be undecidable if we are not careful about the coefficients, and if this has any bearing on the undecidability question.

Reviewer #4 (Remarks to the Author):

Summary:

This paper introduces families of convex semidefinite relaxations for computing the expectation of observables of translationally invariant quantum many bodies in equilibrium, i.e., ground state averages for zero temperature or Gibbs averages for higher temperatures. They consider families of finite or infinite lattices in 1 or 2 dimensions. The algorithms designed in this paper have two important features: (1) the algorithms provide certified upper and lower bounds on the expectation value, and (2) the family converges to the correct bound in the limit. One of the major implications of this result is the decidability of the ground state(or Gibbs state) observable averages (as a promise problem) for the infinite chain (zero temperature for 1D and higher temperature for 2D). This result is significant

because recent literature (e.g., Cubitt, Wolf, and Perez-Garcia) indicates the undecidability of quantities such as spectral gaps or phase diagrams for infinite chains. This result particularly can be viewed as a complementary result to Bausche, Cubitt, Watson (BCW21) where they show a similar problem is undecidable. This apparent paradox is due to the subtle differences between the notion of an infinite system and boundary conditions. In particular, in BCW21, the authors consider an increasing family of finite chains with boundary conditions. In contrast, in this paper, the authors consider an infinite chain without taking the limit (Which, based on my understanding, excludes the boundary condition). The authors explain why the boundary condition in BCW21 is essential in obtaining undecidability. (Based on the authors' claims), this result is the first certified algorithm for general observables in thermal states. The author proves fast convergence for commuting Hamiltonians (1D or high-temperature 2D) and gives quasipolynomial time algorithms to compute the observable expectations in the continuum limit. Their proof is based on notions relevant to the decay of correlation in these systems.

To solve this problem, the authors use novel and nontrivial techniques. To formulate the condition for a state being Gibbs, the author uses the so-called Kubo-Martin-Schwinger (KMS) formulation and includes it as a constraint in their semi-definite program. To use this framework, they have to overcome a few subtle difficulties. First of all, these conditions are formulated using continuous variables. They move from these continuous variable equations to discrete variable families via a matrix representation of the formulation. They then show that this representation implies matrix-valued relative entropy inequalities, which implies well-known energy entropy balance inequalities. These formulations involve difficulties due to the matrix logarithm, as the convex relaxation they consider is no longer an SDP. They use techniques involving SDP approximations of the logarithm. Notably, the authors work out all these formulations in the C^* algebraic framework to apply these techniques to the infinite chains.

The authors also support plausible numerical results and the possibility of application to quantum many-body problems.

I have not verified all the details in the proofs, but I read all of them and found them plausible.

Assessment:

This paper presents a series of important results with novel proof ideas, and it should be accepted.

Comments:

The paper is very well-written, and I don't have extensive comments. As a minor comment, I had a slightly hard time understanding their explanation for the difference between their model and the one considered in BCW21. The authors indicate that the reason why BCW21 obtains undecidability is because they consider specific boundary conditions. Then they say (quote from the paper)

"The definition we use in our work, which is standard in the operator algebraic framework, is more general and in particular contains limits of ground-state observables of finite system Hamiltonians with any choice of boundary conditions."

One may ask, if your algorithm decides the problem for any choice of boundary conditions, why doesn't it decide the specific boundary condition in BCW21? I am not sure why the authors included that sentence. The second minor comment is that the authors may give a more concrete definition of what it means for an algorithm to be certified. I watched the QIP talk, and the presenter initially gave a very simple definition (i.e., it gives upper and lower bounds so that you know how far you are to the correct answer). I think this point can be explained more clearly in the body of the paper. That said, I believe the authors explain everything concretely; my point is about minor presentation improvement.

Dear referees,

We would like to thank all referees for their careful assessment of our manuscript. We found many of their comments extremely useful in clarifying our results and improving its presentation. We hope that the following changes address all points raised.

We submit marked up manuscripts with all changes marked in red. Despite the changes to address the referees' comments, we made some edits to the format. We split the manuscript in a main text containing our main results and an overview of the techniques and the supplementary information containing detailed proofs.

Point-by-point response

Reviewer #1 and #2

1. Infinite ground states and boundary conditions

In Remark 2.1, it is explained that limits of finite volumes ground states produce ground states in the thermodynamic limit, so that different infinite volumes ground states can occur due to different choices of boundary conditions. The latter are defined as extra Hamiltonian terms $\Delta\Lambda_c$ acting on the "external boundary" of a finite volume Λ . Unfortunately the example that follows does not really exemplify what has been just explained. While it contains an example of two different infinite volume ground states, it does not show (at least not explicitly) how these arise from different choices of boundary conditions. We believe making this explicit could be very useful to the readers.

>> We implement this helpful suggestion by making the necessary boundary conditions explicit, showing how to reach the two mentioned ground states by adding open boundary conditions or using an additional term respectively.

Moreover, we think that there are other choices of examples that could have worked better and are more widely known, which we think could be included in addition or substitution of the current one. One example is of course the Ising model, either in 1D or 2D, which is then used as a numerical example in Section 4. Another good example could be Kitaev's toric/planar code, which has single-particle ground states in the limit [CNN18] (which can be obtained by fixing certain boundary conditions).

>> We fully agree that the mentioned examples illustrate the relevance of the concept in current research nicely. Therefore, we added the two examples referring to [CNN18] for details of this explicit construction. We do however also keep our example as it allows a quick back-of-the-envelope demonstration calculation without too involved definitions or solutions.

More generally, since this is a crucial point which separates the current work from the previous results in the literature, this part should be extended and better clarified.

>> We believe that the extended example now containing a fully explicit construction and the illustration through the more common examples address this point.

2. The present algorithm applied to the model on [BCW21]

We think that the discussion at the beginning of section "3.4 Decidability", regarding the relationship between the current work and [BCW21], should be improved and expanded. One question which is

not clear from that section is: what happens if we apply the algorithm presented in this work to the family of models constructed in [BCW21]?

We intuitively believe that the answer should be that the minimum and maximum expectation value of the local observable are always the same, and thus do not depend on whether the particular instance of the problem corresponds to a halting/non-halting machine. Is this interpretation correct? If so, it should be properly explained. Moreover, given the explanation that different infinite volume ground states arise as limits of finite volumes with different boundary conditions, is it possible to explicitly construct the two boundary conditions that realize the minimum/maximum value for these family of models? I.e., can we explicitly construct which boundary conditions we need to impose on the models from [BCW21] that make the problem decidable? Or is the current result showing that such boundary conditions exists, but it could be that constructing them explicitly is also an undecidable problem?

>> We very much appreciate the feedback on this section. Given this comment and also the questions by reviewer #4, we expanded this section. We do not fully know the answer to the questions, but added an explanation of possible scenarios, trying to clarify what can and cannot be the case. We agree with the reviewers interpretation, however, we cannot exclude the additional scenario, that some Turing machines still map to smaller intervals including only 0 or 1, as long as not all possible Turing machines only result in an interval that allows to decide whether the specific Hamiltonian must yield the value 0 or 1 respectively.

>> In our discussion of the Hamiltonian in the reference, we explain how its boundary condition can be changed to invalidate the specific argument. This does not definitely conclude what the outcome of our algorithm would be. In addition, even an explicit construction of this boundary condition does not allow us to use our algorithm to decide the problem in the reference: Our algorithm will simply output the hull of values for these boundary conditions - they cannot be added as an input.

Reviewer #3

1. Can you comment on the applicability of your algorithms to finite-sized but large systems? Do these give any new tools or ideas in that setting?

>> We thank the referee for bringing to our attention that this part is somewhat neglected in our manuscript. Indeed, the formulation of the convex relaxation hierarchy, in particular the reformulation of EEB in terms of the matrix-EEB are novel and relevant for treating finite-size systems. The formulation of our work did not display this fact very prominently so we added an additional comment after Theorem 2.1 to clarify this fact.

2. Page 10, Section 3.2: Can you elaborate why eq. 17e becomes an infinite list of scalar inequalities whereas eq. 17c is not? Related to the previous question, what happens when the system is finite-sized? Can I get a nicer set of inequalities by some kind of polynomial approximation to the logarithm?

>>We added an explanation for the infinite amount of inequalities, see paragraph after equation (11) in the supplementary material: inequality (11e) corresponds to one inequality for each element in the algebra. While the algebra is finite-dimensional, it is still an infinite set. Due to the absence of linearity, reducing this to just basis elements is not straightforward.

>>At the point of Eq. (11) we already reduced to an optimization over a finite sized marginal. This is exactly the same as if we had started from a finite-sized system.

Concerning the approximations to the logarithm, this is in fact what we have used in the decidability result and the numerical experiments. To be precise we used rational approximations because they are more accurate than polynomial approximations and provide upper approximations of the logarithm.

3. Same section, "...outer relaxation for the set of λ -marginals..": Here it may be helpful to emphasise that the relaxation is still optimising over states. This threw me off a little since other outer approximations in the literature do not necessarily optimise over actual states.

>> We added a comment to clarify this in the paragraph after equation (11).

4. I was wondering if SDPs can be undecidable if we are not careful about the coefficients, and if this has any bearing on the undecidability question.

>> The problem of deciding whether the value of an SDP (described by rational coefficients) is bounded by some rational number is decidable exactly via quantifier elimination see [15, 16]. This is what we used in the proof of Theorem 3.7.

Reviewer #4

The paper is very well-written, and I don't have extensive comments. As a minor comment, I had a slightly hard time understanding their explanation for the difference between their model and the one considered in BCW21. The authors indicate that the reason why BCW21 obtains undecidability is because they consider specific boundary conditions. Then they say (quote from the paper)

"The definition we use in our work, which is standard in the operator algebraic framework, is more general and in particular contains limits of ground-state observables of finite system Hamiltonians with any choice of boundary conditions."

One may ask, if your algorithm decides the problem for any choice of boundary conditions, why doesn't it decide the specific boundary condition in BCW21? I am not sure why the authors included that sentence.

>> Our algorithm does not take as input a specific boundary condition; rather the set of all boundary conditions enters the definition of the problem and defines the correct output interval for the problem. Our algorithm then gives outer relaxations of this set. We acknowledge that this point was perhaps not clearly emphasised and we tried to clarify it in the supplemental material Section 2.4, see the response to Reviewer #2, and also in the main text before Definition 2.3.

The second minor comment is that the authors may give a more concrete definition of what it means for an algorithm to be certified. I watched the QIP talk, and the presenter initially gave a very simple definition (i.e., it gives upper and lower bounds so that you know how far you are to the correct answer). I think this point can be explained more clearly in the body of the paper. That said, I believe the authors explain everything concretely; my point is about minor presentation improvement.

>> We changed the paragraph "Certified Algorithms" in the introduction to give a more direct explanation of this concept.

REVIEWERS' COMMENTS

Reviewer #1 (Remarks to the Author):

The authors have addressed our comments in a satisfactory way, and therefore we are happy to recommend the publication of the manuscript as it is.

Reviewer #2 (Remarks to the Author):

Reviewer #3 (Remarks to the Author):

I thank the authors for responding carefully to my comments, in particular comment #2 - I have increased appreciation for the EEB-based relaxations constructed here. The revised version also reads much better. The new additions elaborating on the differences between this work and Ref. 13 are quite helpful. I am happy to recommend acceptance.